# Rapid Amplicon Nanopore Sequencing (RANS) for the Differential Diagnosis of Monkeypox Virus and Other Vesicle-Forming Pathogens

**DOI:** 10.3390/v14081817

**Published:** 2022-08-18

**Authors:** Ofir Israeli, Yehoudit Guedj-Dana, Ohad Shifman, Shirley Lazar, Inbar Cohen-Gihon, Sharon Amit, Ronen Ben-Ami, Nir Paran, Ofir Schuster, Shay Weiss, Anat Zvi, Adi Beth-Din

**Affiliations:** 1Departments of Biochemistry and Molecular Genetics, Israel Institute for Biological Research (IIBR), Ness Ziona 74100, Israel; 2Clinical Microbiology, Sheba Medical Center, Ramat-Gan 52621, Israel; 3Infectious Diseases Unit Tel Aviv Medical Center, Sackler Faculty of Medicine, Tel Aviv University, Ramat Aviv, Tel Aviv P.O. Box 39040, Israel; 4Departments of Infectious Diseases, Israel Institute for Biological Research (IIBR), Ness Ziona 74100, Israel

**Keywords:** Monkeypox virus, Oxford nanopore, Flongle, Orthopoxvirus, vesicle-forming pathogens, variola virus, smallpox

## Abstract

As of July 2022, more than 16,000 laboratory-confirmed monkeypox (MPX) cases have been reported worldwide. Until recently, MPX was a rare viral disease seldom detected outside Africa. MPX virus (MPXV) belongs to the Orthopoxvirus (OPV) genus and is a genetically close relative of the Variola virus (the causative agent of smallpox). Following the eradication of smallpox, there was a significant decrease in smallpox-related morbidity and the population’s immunity to other OPV-related diseases such as MPX. In parallel, there was a need for differential diagnosis between the different OPVs’ clinical manifestations and diseases with similar symptoms (i.e., chickenpox, herpes simplex). The current study aimed to provide a rapid genetic-based diagnostic tool for accurate and specific identification of MPXV and additional related vesicle-forming pathogens. We initially assembled a list of 14 relevant viral pathogens, causing infectious diseases associated with vesicles, prone to be misdiagnosed as MPX. Next, we developed an approach that we termed rapid amplicon nanopore sequencing (RANS). The RANS approach uses diagnostic regions that harbor high homology in their boundaries and internal diagnostic SNPs that, when sequenced, aid the discrimination of each pathogen within a group. During a multiplex PCR amplification, a dA tail and a 5′-phosphonate were simultaneously added, thus making the PCR product ligation ready for nanopore sequencing. Following rapid sequencing (a few minutes), the reads were compared to a reference database and the nearest strain was identified. We first tested our approach using samples of known viruses cultured in cell lines. All the samples were identified correctly and swiftly. Next, we examined a variety of clinical samples from the 2022 MPX outbreak. Our RANS approach identified correctly all the PCR-positive MPXV samples and mapped them to strains that were sequenced during the 2022 outbreak. For the subset of samples that were negative for MPXV by PCR, we obtained definite results, identifying other vesicle-forming viruses: Human herpesvirus 3, Human herpesvirus 2, and Molluscum contagiosum virus. This work was a proof-of-concept study, demonstrating the potential of the RANS approach for rapid and discriminatory identification of a panel of closely related pathogens. The simplicity and affordability of our approach makes it straightforward to implement in any genetics lab. Moreover, other differential diagnostics panels might benefit from the implementation of the RANS approach into their diagnostics pipelines.

## 1. Introduction

As of July 2022, more than 16,000 laboratory-confirmed monkeypox (MPX) cases, including one death, have been reported to WHO across more than 40 countries. The outbreak of MPX continues to primarily affect men who have reported recent sex with new or multiple male partners [1]. Until recently, MPX was a rare viral disease seldom detected outside Africa [2,3], and the reasons for the rapid and global emergence of MPX are still elusive. MPX virus belongs to the Orthopoxvirus (OPV) genus in the Poxviridae family that encompasses over 70 members of large DNA genome viruses (roughly 200,000 bp) that encode about 200 proteins and replicate in the cytoplasm of infected cells. This family is subdivided into two subfamilies, the Entomopoxvirinae and the Chordopoxvirinae, which infect insects and vertebrates, respectively. The Chordopoxvirinae have been further subdivided into eight genera (OPV, Avipoxvirus, Capripoxvirus, Leporipoxvirus, Molluscipoxvirus, Parapoxvirus, Suipoxvirus, and Yatapoxvirus). Within the OPV genus, only five can infect humans: Variola virus, MPX virus, Vaccinia virus, Cowpox virus, and Camelpox virus [4,5,6,7], mainly via the airway’s epithelium through an aerosol or via the skin. Variola virus, a member of the OPV genus, is the causative agent of smallpox, which is a major health and biosecurity concern. Smallpox is a highly contagious human-specific disease with excessive mortality of up to 40% in unvaccinated populations. It is estimated that smallpox disease has killed approximately 300–500 million people during the 20th century [6]. Smallpox is the first viral disease in human history to have been eradicated. In 1980, the World Health Organization (WHO) certified the eradication of the Variola virus following a successful worldwide vaccination campaign. Since then, the only known smallpox virus samples have been kept in specific repositories in Russia and the USA [5,8,9].

Following the eradication of smallpox, the vaccination campaign ceased and the population immunity against smallpox and other OPV-related diseases has significantly decreased. Since then, sporadic cases of MPX cases have been reported, mainly in Africa [2,3]. Differential diagnosis between viruses causing rash and vesicular lesions is sometimes challenging, mainly upon the emergence of pathogens in non-endemic areas. The current MPX outbreak indeed challenges the differential diagnosis capabilities and emphasizes the importance of the diagnosis of OPVs and other vesicle-forming pathogens. The recombination potential of OPVs, in conjunction with an exceptional ability to cross animal species barriers, substantially increases their pathogenic potential [10]. The development of a reliable and rapid method for sensitive detection of MPXV and accurate discrimination between the former and other OPV members or additional vesicle-forming pathogens is of immense importance. The inaccessibility of the Variola virus and the high level of genetic similarity within the OPV group (over 95% similarity at the DNA level) [11,12] restricts the ability to develop a reliable diagnostic test. Several approaches have been utilized for the identification of MPXV and differentiation between OPV members, including typical growth in chorioallantoic membranes, serology, and mapping with DNA restriction enzymes [13,14]. Recently, Luciani et al. [15] developed a real-time PCR assay targeting a highly conserved region of the poxvirus genome, thus allowing a sensitive pan-poxvirus detection. However, this method is lacking in the differential diagnosis potential for viruses within the OPV genus.

The most specific MPXV detection is based on real-time PCR assays. The tests developed by Li et al. and Maksyutov et al. [16,17] can diagnose MPXV and distinguish between the West African clade and the Central African clade. This differential diagnosis is important; as the Central African strains’ illness is typically accompanied by symptoms similar to discrete, ordinary smallpox and has a case fatality rate of approximately 10% in unvaccinated populations, whereas the West African strains appear to cause a less severe disease [16].

Several genetic detection approaches were developed for rapid discrimination of OPV members, including conventional PCR assays with or without amplicon RFLP analyses, real-time PCR tests, and PCR followed by cross-hybridization tests that use dot-blot or microarray for differential analysis [18,19,20,21,22]. These techniques can discriminate between known human infectious OPVs (Vaccinia, Variola, Cowpox, or MPX) to a reliable extent, but usually fail to pinpoint the vesicle-forming pathogen in a suspected sample if it is not from the OPV group. Moreover, the above-mentioned methods cannot discriminate between clades within the same species, which might be important in a new outbreak. High-throughput sequencing (HTS) is currently the gold standard technique for optimal differential genomic discrimination [23], but the feasibility of using HTS-based methods for the differential diagnosis of vesicle-forming viruses is hampered by the high, mainly human, genomic background in clinical samples, raising the need for virus isolation and propagation, which is time-consuming and labor intensive. In the past, we addressed this challenge by utilizing a microarray-based detection system, which was based on an arrayed primer extension approach, for the detection of differential single nucleotide polymorphisms (SNPs). Although this system is reliable, it is also labor- and time-intensive, outdated, and costly [24].

The current study aimed to provide a rapid diagnostic tool for accurate and specific genetic detection of MPXV and closely related species, as well as other vesicle-forming pathogens, which might lead to misdiagnosis. We developed an approach that we termed rapid amplicon nanopore sequencing (RANS) that is based on Oxford Nanopore sequencing technology. This approach is highly accurate, rapid, and low-priced. The approach is presented in Figure 1 and elaborated in Section 2.1.

For our RANS-based detection of vesicle-forming pathogens, we initially assembled a list of relevant pathogens. The list included 14 viral species, all of which have full-length genomic sequences in the databases, causing infectious diseases associated with vesicles, prone to be misdiagnosed as MPX. The pathogens in the list were chosen according to the following criteria: clinical features and the resemblance of symptoms (vesicle formation), incidence of the disease, severity of the disease in humans, mortality rate, and human infectivity. Naturally, five members of the OPV genus that are genetically and morphologically similar to MPX were included in the list (Table 1). The 14 pathogens belong to two main subfamilies: Chordopoxvirinae and Alphaherpesvirus. The pathogens within each group are closely related and share a high degree of sequence similarity. In a previous work [24], we found nine potential differential diagnostic regions that are detailed in Table 1. These regions contain sequence homology in their termini that enable the design and use of common primers for multiplex PCR amplification of the diagnostic region from all related pathogens in a particular subgroup. The amplified fragment harbors diagnostic SNPs that can be utilized for the specific identification of each pathogen.

## 2. Materials and Methods

### 2.1. Cell Lines and Virus-Infected Samples

Vero cells (CCL-81, ATCC) were infected with the viruses listed below as previously described [25]. Following 48 h of incubation, cells and culture media were collected, and total DNA was extracted using the QIAamp DNA mini kit (Qiagen, Hilden, Germany). Vaccinia virus (VACV) lister, Vaccinia virus Western Reserve (VACV-WR), Ectromelia virus (ECTV) Moscow, and Cowpox virus (CPXV) Brighton-red were previously described [26,27]. Human herpesviruses 1 and 3 were from the Israel Institute for Biological Research (IIBR, Ness Ziona, Israel) virus collection. All viruses were propagated on Vero cells as described [28].

### 2.2. Clinical Samples

During the 2022 MPX outbreak, our diagnostic lab at IIBR received a variety of clinical samples that were suspected by physicians to contain MPXV. The IIBR is the Israeli authorized national reference laboratory to process the samples for the identification and characterization of MPXV. All samples subjected to viral genetic characterization were processed in an anonymized fashion. The amplicons produced were strictly designed for the amplification of vesicle forming viruses for RT-PCR or Oxford Nanopore sequencing downstream analysis.

### 2.3. DNA Extraction

DNA was extracted from the samples using the QIAamp DNA Mini Kit (Qiagen) with a protocol for blood and body fluids in a QIAcube robot and was eluted in 100 μL H_2_O.

### 2.4. Multiplex PCR

Multiplex PCR was performed using a two-step PCR strategy: a PCR product was at first generated using specific primers flanked by a multiplex 5′ phosphorylated adapter sequence (Table 2 in bold) and then further amplified in the second reaction by the adapter sequence. The PCR was performed using the Qiagen multiplex PCR kit (Qiagen) in a final volume of 25 µL using two separate primer mixes (50 nM each primer, except for Y71 primers, 150 nM each) A or B (see Table 2 for details).

A total of 5 µM of the multiplex 5′ phosphorylated adapter was added to each mix. The PCR was carried out under the following conditions: 15 min at 95 °C, and then 10 cycles for 30 s at 94 °C, 90 s at 57 °C, and 90 s at 72 °C followed by 32 cycles for 30 s at 94 °C, 90 s at 57 °C, and 90 s at 72 °C and a final extension 15 min at 72 °C.

### 2.5. Amplicon Library Preparation

The PCR amplicons from the previous step were purified using 0.8× AMPure XP beads (Beckman, Indianapolis, IN, USA) (final elution in 50 µL H_2_O), quantified using a Qubit HS DNA kit (Thermo scientific, Waltham, MA, USA), and diluted to 1 ng/µL. A total of 30 µL of the diluted amplicons were added to 12.5 µL Ligation Buffer (LNB), 5 µL NEBNext Quick T4 DNA Ligase, and 2.5 µL Adapter Mix (AMX) (all from Oxford Nanopore technologies—ONT). The ligation was performed at RT for 10 min. Following the ligation, the amplicons were purified again with 0.8× AMPure XP beads, with a final elution in 10 µL EB.

### 2.6. Flongle Loading and Sequencing

The Flongle was pre-washed with a wash mix (117 µL of Flush Buffer with 3 µL of Flush Tether) and loaded with the sequencing mix containing 15 µL Sequencing Buffer II (SBII), 10 µL Loading Beads II (LBII)), and 5 µL of the ligation product following the gDNA-sqk-lsk109-GDE_9063 protocol (All materials from ONT).

### 2.7. Analysis

An in-house database representing vesicle-forming pathogens was constructed. This database comprised 1044 complete genomes from the Herpesviridae and Poxviridae families, downloaded from the Virus Pathogen Resource (ViPR) website [29]. The database was further updated to include 75 monkeypox variants from the current 2022 outbreak, downloaded from the NCBI repository (www.ncbi.nlm.nih.gov, accessed on 18 July 2022) ). The complete list of strains in our database is detailed in Appendix A. All reads generated by the Flongles were aligned with minimap2 [30]—a versatile sequence alignment program for long DNA or mRNA sequences, against the constructed database. The reads were mapped with the parameter-ax map-ont for ONT reads in the output format SAM. For viewing, sorting, and indexing the SAM file, the Samtools package [31] was performed with the parameters view-Sb, sort, and index, respectively.

### 2.8. MPXV Real-Time PCR

Multiplex real-time PCR assays were performed in a 50 μL reaction volume using the SensiFAST™ Probe Lo-ROX kit (BIOLINE). The mix contained viral-specific primers (30 pmol per reaction each) and probes (15 pmol per reaction each) detailed below:

MPXV generic assay (GE)

forward primer (5′-GGAAAATGTAAAGACAACGAATACAG)

reverse primer (5′-GCTATCACATAATCTGGAAGCGTA)

probe (5′Joe-AAGCCGTAATCTATGTTGTCTATCGTGTCC-3′BHQ1)

MPXV West African specific assay (WA)

forward primer (5′-CACACCGTCTCTTCCACAGA)

reverse primer (5′-GATACAGGTTAATTTCCACATCG)

probe (5′FAM-AACCCGTCGTAACCAGCAATACATTT-3′BHQ1)

The PCR was carried out on a QuantStudio 5 real-time PCR system (Applied Biosystems), under the following conditions: 20 s at 95 °C followed by 40 cycles at 1 s 95 °C and 20 s 60 °C [16].

## 3. Results and Discussion

### 3.1. Establishment of the Rapid Amplicon Nanopore Sequencing (RANS) Approach for the Differential Detection of MPXV and Other Vesicle-Forming Pathogens

Using in-house bioinformatical tools, we previously identified diagnostic regions derived from common genes or sequences for the detection and differentiation of vesicle-forming pathogens (Table 1, [24]). As the assay was based on outdated microarray technology, we wanted to utilize modern HTS to achieve similar capabilities. In this study, we developed a rapid and accurate differential diagnosis approach, named RANS (rapid amplicon nanopore sequencing) (Figure 1) based on ONT-based sequencing technologies. A RANS-compatible diagnostic region consists of sequences that harbor high homology in its boundaries that enables the use of common primers for PCR amplification of the region from all related pathogens in a particular subgroup without the downside effects of primer abundance on multiplex PCR reactions. The center of the fragment contains diagnostic SNPs that enable the specific identification of each pathogen within a group. During the multiplex PCR amplification, a 5′ phosphate (using a phosphorylated primer) and a dA tail (using the terminal transferase activity of the PCR DNA polymerase) are simultaneously added, thus making the PCR products ready for ligation with the Oxford Nanopore adapters. These adapters facilitate strand capture and loading of a processive enzyme (motor enzyme) at the 5′-end of one strand. The enzyme is required to ensure unidirectional single-nucleotide displacement along the strand at a millisecond time scale. The adapters also concentrate DNA substrates at the membrane surface proximal to the nanopore, boosting the DNA capture rate by several thousand-fold [32]. After a short ligation to the adapters (10 min), the amplicon libraries are ready for nanopore sequencing. Subsequently, the amplicons are sequenced on an Oxford Nanopore MinION Flongle device that enables direct, real-time, and relatively low-cost DNA sequencing, utilizing single-use flow cells (see the Materials and Methods for details and Figure 1). The readout sequences are next mapped to the in-house Herpesviridae and Poxviridae designated database (see the Materials and Methods for the database details) using minimap2. For each sample, the strain that gained the highest percentage of mapped reads was recorded as the nearest detected strain. The ONT Flongles sequenced our amplicons at a rate of 200–600 sequences per minute. As we and others found that ≈1000 mapped sequences are required for accurate identification of a pathogen [33,34,35], a few-minute run was sufficient for data collection. Our RANS approach for the rapid identification of vesicle-forming pathogens is summarized in Figure 1.

### 3.2. Laboratory Samples

We initially examined our RANS diagnostic approach using eight different vesicle-forming viruses, cultured on Vero cell lines, of which six are OPV members (samples 1–6, Table 3). As presented in Table 3, the application of the RANS approach resulted in the detection and correct identification of the virus species for each of the eight samples examined. Moreover, identification at the strain level is also exemplified, as in the case of the Vaccinia virus. Overall, the analysis resulted in more than 90% identity (range 91.0–99.1%) to a specific strain in our database, of 1119 strains (detailed in Appendix A). As few as 9000 mapped reads allowed for the identification of sample 4 CMLP, identified as Camelpox Negev 2016 and at 98.4%.

The different viruses’ abbreviations are detailed in Table 1 legend.

### 3.3. Evaluation of RANS in Clinical Samples

Following the successful identification of viruses that were grown in cell cultures, we sought the identification of more challenging clinical samples. During the 2022 MPX outbreak, our diagnostic lab at IIBR received a variety of clinical samples that were suspected by physicians to contain MPXV. All the suspected samples were analyzed by first-line real-time PCR assays, which are highly specific and sensitive for the identification of MPXV [16]. Of nine analyzed samples, all exhibiting the common chrematistics of vesicles and therefore diagnosed by the physician as possible MPXV, four were positive for MPXV, while the other five were negative. The Cts of the positive samples ranged from 24.7 to 38.8, implying moderate to very low viral concentration in the positive MPXV samples. All nine samples were subsequently subjected to analysis by our RANS approach. The results are summarized in Table 4. Our RANS approach correctly identified the four MPXV PCR-positive samples. The diagnosis was robust, swift, and accurate also for very low viral loads. For example, sample #2043 with 38.8 and 38.5 Cts (for the GE and WA real-time PCR tests, respectively), which corresponds to 10–100 genome equivalents/mL (data not shown), showed 99.8% identity to MPXV. As expected, all four MPXV-positive samples were mapped to the Monkeypox virus in the in-house constructed database. When the sequenced reads were compared to a database updated with MPXV 2022 outbreak sequences (a total of 75 sequences available in the NCBI repository as of June 2022), the four MPXV-positive samples were assigned to sequences originating from the current outbreak and belonging to the same clade [36]. Of the five PCR-negative MPX samples, four were identified as other vesicle-forming pathogens, while one sample (#2114) did not reveal any known relative in the database representing the 14 vesicle-forming pathogens. Overall, the RANS approach exhibited differential diagnosis capabilities at the utmost level in diverse and challenging clinical samples.

## 4. Summary and Conclusions

This work is a proof-of-concept study, describing the establishment of the RANS approach and demonstrating its potential for discriminatory identification of closely related pathogens. Using this approach, we here demonstrate the rapid discrimination between a panel of 14 vesicle-forming viruses, among them MPXV and closely related OPV members (VACV, CPXV, and ECTV) during the recent spread of the 2022 MPX outbreak. The simplicity of our RANS approach makes it straightforward to implement in any genetics lab, using a standard PCR machine and ONT sequencer with a disposable Flongle, which costs under USD 100. RANS could potentially distinguish between viruses’ clades, subtypes, and strains up to an ultimate single base resolution. Other differential diagnostics panels could benefit from the implementation of the RANS approach into their diagnostics pipelines.

## Figures and Tables

**Figure 1 viruses-14-01817-f001:**
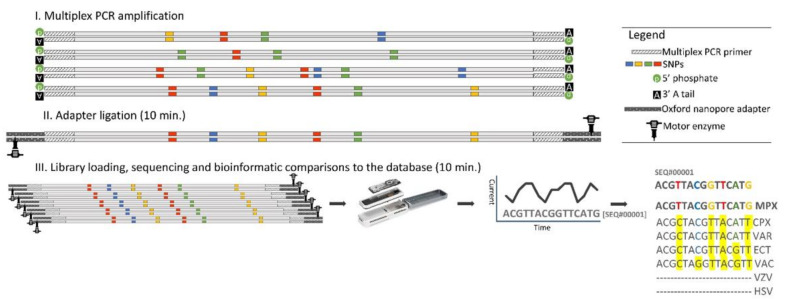
The RANS approach for the differential diagnosis of MPXV and other vesicle-forming pathogens. The figure elaborates on the rapid and comprehensive RANS procedure. The procedure starts with a suspected clinical sample and ends with a specific sequence-based identification in a timeframe of a few hours. (I). Multiplex PCR amplification, in which primers that are mutual for a group of viruses (for example OPVs) are used to amplify the whole group. A 5′ phosphate (using a phosphorylated primer) and a dA tail (using the terminal transferase activity of the DNA polymerase and a final extension step of 15 min) were concurrently added at this stage. Next (II), an adapter ligation was performed, wherein the ONT-proprietary adapters, which were attached to a motor enzyme, ligated to the ends of the strands. The final step (III) involved library loading, sequencing, and base calling: The amplicons were loaded onto a Flongle apparatus and based-called FASTQ files were generated as the DNA was translocated through the nanopores. Finally, bioinformatic comparisons between the resulting sequences to a relevant database pinpointed the specific agent (in bold). The different viruses’ abbreviations are detailed in Table 1 legend.

**Table 1 viruses-14-01817-t001:** Vesicle-forming pathogens and their diagnostic regions in the RANS assay.

Subfamily		*Chordopoxvirinae*										*Alphaherpesvirus*			
Genus		*Orthopoxvirus*						*Parapoxvirus*		Moll.	Yata.	Simplexvirus			Varic.
**Area**	**bp**	**VARV**	**MPXV**	**VACV**	**CPXV**	**CMLP**	**ECTV**	**ORF**	**BPSV**	**MOLL**	**TANA**	**HSV1**	**HSV2**	**HBV**	**VZV**
**Chem**	597	√	√		√ √	√	√								
**R60**	638	√ √	√ √	√ √	√ √	√ √	√ √								
**R56**	652	√ √	√ √	√ √	√ √	√	√								
**Y71**	785	√	√	√	√ √		√				√ √				
**m8121L**	720	√ √	√	√	√	√ √	√ √				√ √				
**Mine**	1017	√ √	√		√ √	√	√ √	√ √	√ √	√ √					
**B100**	355							√ √	√ √	√ √					
**ORF45**	530											√ √	√ √	√ √	√ √
**ORF27**	460											√ √	√ √	√ √	√ √

Nine pre-selected diagnostic regions harboring specific SNPs for identification of 14 pathogens belonging to Chordopoxvirinae and Alphaherpesvirus subfamilies (√√ = more than 10 potential single nucleotide polymorphisms (SNPs); √ = 1–10 potential SNPs; no mark = no potential SNPs). The selection of the regions was described in [24]. Abbreviations in the table: base pairs, bp; Molluscipoxvirus, Moll.; Yatapoxvirus, Yata; Varicellovirus, Varic.; Variola virus, VARV; Monkeypox virus, MPXV; Vaccinia virus, VACV; Cowpox virus, CPXV; Camelpox virus, CMLP; Ectromelia virus, ECTV; Orf virus, ORF; Bovine papular stomatitis virus, BPSV; Molluscum contagiosum virus, MOLL; Tanapox virus, TANA, Herpes simplex virus type 1, HSV1; Herpes simplex virus type 2, HSV2; Herpes B virus, HBV, Varicella-zoster virus (i.e., Human herpesvirus 3), VZV.

**Table 2 viruses-14-01817-t002:** Primers for the diagnostic regions in the RANS assay.

Mix	Primer	Sequence
**A**	Mine-sF	**CGATACGACGGGCGTACTAGCG**AGGTGCTCwGCGAGAAGTTCAC
	Mine-sR1	**CGATACGACGGGCGTACTAGCG**GGCAGCACCAGCATGAACTTG
	Mine-sR2	**CGATACGACGGGCGTACTAGCG**GGCAGCATGAGCATGAACTTG
	Y71-F	**CGATACGACGGGCGTACTAGCG**CCCGTwTATGGATCwATTCAAGA
	Y71-R	**CGATACGACGGGCGTACTAGCG**CCTCTTCCyTCyGGATCCTTAGA
	R60-F	**CGATACGACGGGCGTACTAGCG**CAATATGAACAAGAAATAGAATCGTTAGAAG
	R60-R	**CGATACGACGGGCGTACTAGCG**GCGCTTCTATATCTCTCATTAGCTAGAA
	ORF45-F	**CGATACGACGGGCGTACTAGCG**TGCGAyGAyCACATGCCG
	ORF45-R	**CGATACGACGGGCGTACTAGCG**TCCTGGCTGCTrTTkCCCTC
**B**	Chem-F	**CGATACGACGGGCGTACTAGCG**CAAmCATyATATGGGAATCGATGTTA
	Chem-R	**CGATACGACGGGCGTACTAGCG**TCGAyATACTTrAATCCATCCTTGAC
	ORF27-F1	**CGATACGACGGGCGTACTAGCG**CGTGGCCGACAACTGCCT
	ORF27-F2	**CGATACGACGGGCGTACTAGCG**ACAGTGGCGGATAACTGCCT
	ORF27-R1	**CGATACGACGGGCGTACTAGCG**CGTGCTCTGCCACACGTG
	ORF27-R2	**CGATACGACGGGCGTACTAGCG**CGTGCTCTGCCACACGTG
	B100-mF	**CGATACGACGGGCGGACTAGCG**GTACCACCCCAGCCAGTACG
	B100-R	**CGATACGACGGGCGTACTAGCG**AyGCGCACCTCGTTCAT
	R56-F	**CGATACGACGGGCGTACTAGCG**TTACCACGTCTGGATAGGAGATTGT
	R56-R	**CGATACGACGGGCGTACTAGCG**CGTGTTCTTAGTTGCTTAGCTGAAAC
	m8121-sF1	**CGATACGACGGGCGTACTAGCG**TACTTGGAAAAGAATTTGGACCAA
	m8121-sF2	**CGATACGACGGGCGTACTAGCG**GAGAAATTCCATTTAATATGAAAGACATG
	m8121-sR1	**CGATACGACGGGCGTACTAGCG**CATGATAATATTAAAGATAAAGCGCTGAC
	m8121-sR2	**CGATACGACGGGCGTACTAGCG**GGTTCGCTTAAAAATATAGACTTGTTAAATG
**Multiplex adapter**	P* **CGATACGACGGGCGTACTAGCG**	

In bold, the adapter sequence; in black, the specific primers for each diagnostic region; P*, phosphorylation at the 5′ of the adapter.

**Table 3 viruses-14-01817-t003:** RANS laboratory samples’ results.

Sample #	Virus Strain Cultured	% Mapped Reads	Species Detected	Nearest Detected Strain (Accession Number)
1 VACV	Vaccinia Lister	94.9	Vaccinia virus	Vaccinia Lister (KX061501)
2 VACV	Vaccinia Western Reserve	93.1	Vaccinia virus	Vaccinia Western Reserve (AY243312)
3 ECTV	Ectromelia Moscow	91.0	Ectromelia virus	Ectromelia Moscow (AF012825)
4 CMLP	Camelpox Negev 2016	98.4	Camelpox virus	Camelpox Negev 2016 (MK901851)
5 CPXV	Cowpox Brighton Red	97.0	Cowpox virus	Cowpox Brighton Red (AF482758)
6 MPXV	Monkeypox virus Israel	99.1	Monkeypox virus	Monkeypox virus Israel 2018 (MN648051)
7 VZV	Human herpesvirus 3	96.8	Human herpesvirus 3	Human herpesvirus 3 YCO1 (KU926318)
8 HSV1	Human herpesvirus 1	93.8	Human herpesvirus 1	Mckrae (MN136524) Human herpesvirus 1

**Table 4 viruses-14-01817-t004:** RANS clinical samples’ results.

Clinical Sample #	Source	MPXV Ct GE WA		% Mapped Reads	Species Detected	Nearest Strain (Accession Number)
2022	vesicle swab	24.7	25.0	99.6	Monkeypox virus	MPXV/Germany/2022/RKI015 (ON694331.1)
2023	throat swab	32.2	32.7	99.6	Monkeypox virus	MPXV/Germany/2022/RKI020 (ON694337.1)
2043	throat swab	38.8	38.5	99.8	Monkeypox virus	MPXV/Germany/2022/RKI03 (ON682263.2)
2072	semen	34.1	33.7	99.5	Monkeypox virus	MpxV/Spain/MD-HGUGM-6532064/2022 (ON720849)
2012	vesicle swab	undet.	undet.	95.0	Human herpesvirus 3	YC02 (KU926319)
2068	vesicle swab	undet.	undet.	99.9	Human herpesvirus 3	YC02 (KU926319)
2089	vesicle swab	undet.	undet.	89.6	Human herpesvirus 2	HG52 (Z86099)
2090	vesicle swab	undet.	undet.	97.9	Molluscum contagiosum virus subtype 1	MCV1_P05S02A (MN931749)
2114	vesicle swab	undet.	undet.	0.0	unknown	unknown

The different viruses’ abbreviations are detailed in Table 1 legend. undet. = undetected; GE, MPXV generic real-time PCR assay [16]; WA, West African specific real-time PCR assay [16].

## Data Availability

All the sequenced amplicons from the 16 samples that were analyzed in this study were deposited into the Sequence Read Archive (SRA) at NCBI as fastq files (BioProject ID PRJNA860737). The deposited file names of the different samples correspond to the names in Table 3 and Table 4 left columns.

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
