# Peer review of "Rapid Amplicon Nanopore Sequencing (RANS) for the Differential Diagnosis of Monkeypox Virus and Other Vesicle-Forming Pathogens"

_viruses, 2022, doi:10.3390/v14081817_

Round 1

Reviewer 1 Report

For OPXVs, viral sequences share a high level of similarity. Therefore, to distinguish different strains of OPXVs by traditional qPCR method is restricted by few choices of primers and probes. To the best of my knowledge, only two qPCR primer sets could be used to distinguish the two clades of MPXV. In this study, the authors employed the Nanopore sequencing technology to distinguish MPXV strains. Their method abolishes the need to design a strain-specific qPCR primer set, and distinguishes different MPXV strains by direct sequencing of viral amplicons containing strain-specific SNPs.

Minor concerns:

1. Line 54-55. Camelpox virus can also spill over to cause zoonotic infection in humans (doi: 10.1016/j.vetmic.2011.04.010)

2. Line, 98-99. As stated by the authors, Li et al. and Maksyutov et al. developed qPCR method to distinguish MPXV between the West African clade and the Central African clade.

3. Line 115. The assembled pathogens are full-length viral genome or diagnostic fragments?

4. Line 256, the procedure.....The procedure

Reviewer 2 Report

This manuscript describes a rapid genetic-based diagnostic tool for identification of vesicle-forming pathogens including monkeypox. The method uses nanopore sequencing that using PCR primers that have high homology to the boundaries within a virus group and internal diagnostic SNPs revealed by sequencing. The entire procedure takes only a few hours and can be done in the field. The authors validate the assay with both laboratory and clinical samples. The data are good and the paper well written.

Author Response

Dear reviewer,

We would like to thank you for your interest in the manuscript and for your fruitful examination. It was a great pleasure for us to receive your professional opinion on our work.

Reviewer 3 Report

General comments:

This article presents a new and rapid sequencing method for differential diagnosis of Monkeypox virus and other vesicle-forming pathogens. The authors have used two multiplex PCR primers sets to amplify 10 poxviruses and 4 herpesviruses (previously developed for another application), followed by processing through Nanopore sequencing. Rapid and accurate diagnosis is indeed essential for Monkeypox infections and this technique allowed for confirmation of MPXV in 4 clinical samples, and correct identification of previously undetermined pathogens in 4 out of 5 other samples. 

- However, in view of table 4, qPCR was sufficient, faster (and cheaper) to detect MPXV in tested clinical samples and though correct diagnosis if of major importance, the authors’ RANS approach “only” allowed identification on otherwise undetermined samples. 

Compared to commonly used qPCR amplification (GE and WA), how would this new technique cope with new MPXV variants?

- Nanopore sequencing has already been used for direct identification on DNA extracted from Avipoxvirus lesion (without PCR amplification) (see Croville et al, J Virol Methods, 2018 Nov; 261:34-39. doi: 10.1016/j.jviromet.2018.08.003), which would appear to be faster and sufficient for direct MPXV diagnosis. In this case, what would be the advantages of a differential diagnosis such as the one the authors are presenting? 

- Table 1: Are the 1-10 or >10 potential SNPs referring to SNP within a species, thus allowing strain identification? In that case what is the identity percentage between close species (among orthopoxviruses for example) for each targeted sequence? (I understand it was presented in a previous publication but I cannot access its content)

- Tables 3 and 4: what were the minimum number of reads and mapped reads necessary for each virus? Line 252, a minimum of 1,000 mapped sequences is mentioned, but, line 272, 9,000 mapped reads were “sufficient” for sample 4, so to include this data in the tables would be useful.

- Though herpesviruses primers include HSV1 and HSV2, but not HSV3 sets, the % mapped reads were better for samples 2012 and 2068 (HSV3) than for sample 2089 (HSV2). DO the authors have any idea on this?

4- More specific comments:

- Line 50 “…in the cytoplasm.” Please add “in the cytoplasm of infected cells”

- Lines 59-60 “it is estimated that smallpox…” a reference is required her

- Line 146 “All viruses were propagated on Vero cells as described”. The corresponding reference is missing
